# Portuguese Lipid Study (e_LIPID)

**DOI:** 10.3390/jcm13226965

**Published:** 2024-11-19

**Authors:** Joana Rita Chora, Ana Catarina Alves, Cibelle Mariano, Quitéria Rato, Marília Antunes, Mafalda Bourbon

**Affiliations:** 1Grupo de Investigação Cardiovascular, Departamento de Promoção da Saúde e Prevenção de Doenças não Transmissíveis, Instituto Nacional de Saúde Doutor Ricardo Jorge, 1649-016 Lisbon, Portugal; 2BioISI, Biosystems & Integrative Sciences Institute, Faculdade de Ciências, Universidade de Lisboa, 1749-016 Lisbon, Portugal; 3Serviço de Cardiologia, Unidade Local de Saúde da Arrábida, EPE, 2910-446 Setúbal, Portugal; 4CEAUL—Centro de Estatística e Aplicações, Faculdade de Ciências, Universidade de Lisboa, 1749-016 Lisboa, Portugal

**Keywords:** cardiovascular risk factors, dyslipidaemia, lipid profile, lipid percentiles

## Abstract

**Background/Objectives**: Incidence of cardiovascular disease (CVD) is increasing in low- and middle-income countries because of changing lifestyles. Since dyslipidaemia is a major independent cardiovascular risk factor, its correct identification is critical to implement specific interventions for CVD prevention. This study aimed to characterise the lipid profile of the Portuguese population. **Methods**: Overall, 1688 individuals from the general population (e_COR study, 2012–2014) were included. Population-specific percentiles for ten lipid biomarkers were estimated by bootstrapping methods to ensure national representativity. Statistical analyses were performed using RStudio. **Results**: The 50th percentile estimated for total cholesterol (TC), LDL-C, and non-HDL-C are similar to scientific societies recommended values for the general (low or moderate risk) population. National prevalence of having lipid parameters above recommended values was 64.6%, 66.9%, 51.3%, 68.9%, 17.8%, and 21.1% for TC, LDL-C, apoB, non-HDL-C, triglycerides, and Lp(a), respectively; these values are generally higher in men and increasing with age, except for Lp(a). A high prevalence of severe dyslipidaemia (>90th percentile) was identified, highest for small dense LDL-C (31.3%), apoB (30.4%), and LDL-C (30.3%). The national prevalence of CVD events was 5%. Three individuals were genetically identified with familial hypercholesterolemia, a high CVD risk condition. **Conclusions**: We provide for the first-time lipid biomarker percentiles for the general Portuguese population. Our results highlight that hypercholesterolemia is a neglected cardiovascular risk factor with over half of the population with TC, LDL-C, and apoB above recommended values. Since hypercholesterolemia is a modifiable risk factor, strategies to increase adherence to changes in lifestyle habits and medication need to be urgently discussed.

## 1. Introduction

Cardiovascular disease (CVD) is the leading cause of death among non-communicable diseases and disabilities worldwide, affecting both developed and developing countries. It accounts for 47% of deaths in Europe [1,2]. The aetiology of CVD is multifactorial, with several risk factors that are potentially modifiable. Dyslipidaemia, hypertension, and cigarette smoking are three well-known major risk factors for CVD that can be controlled. Effective management of these major cardiovascular risk factors has been shown to significantly reduce the risk of CVD [3].

Epidemiological studies have established a link between CVD and elevated plasma lipid levels, such as total cholesterol (TC), non-high-density lipoprotein cholesterol (non-HDL-C), low-density lipoprotein cholesterol (LDL-C), and triglycerides (TG). Conversely, a low concentration of high-density lipoprotein cholesterol (HDL-C) is also associated with CVD. Individually and collectively, these lipid changes contribute to the development of atherosclerosis [4].

Extensive evidence from large-scale prospective studies has demonstrated that LDL-C lowering therapies, primarily statins, significantly reduce the risk of CVD events in high-risk patients. For every 1 mmol/L (39 mg/dL) decrease in LDL-C, the risk of major cardiovascular events is reduced by 21% [5].

The risk factor profile can vary across populations due to differences in nationality, ethnicity, genetics, socio-cultural, and economic factors. Therefore, understanding the specific risk factor profile of each population is crucial. As the population ages, periodic assessment of the prevalence of cardiovascular risk factors is necessary. This information will help predict cardiovascular mortality trends in the coming years and assist in designing preventive strategies to address this significant health issue [6,7].

It is equally important to establish population-specific age and sex reference intervals for better interpretation of clinical laboratory tests and patient care. Biomarker percentiles can be used to define reference intervals for normal and pathological values, providing the relative position of an individual within a population. Percentile calculations are advantageous because they are not strongly influenced by extreme values, unlike mean values, and do not require normally distributed data, making them applicable even for skewed data [8].

Percentiles can be obtained using various methods, including bootstrap methods, which are increasingly used in medical literature, especially for non-Gaussian population distribution or when the distribution is unknown. In a bootstrap, a dataset is randomly resampled with replacement multiple times, and statistical conclusions are drawn from these resamples. Although the nonparametric bootstrap is a very computer-intensive method, it is valuable for determining confidence intervals (CI) of a quantile (e.g., 0.05 to 0.95) or percentile (e.g., 5th to 95th) [9,10].

The aim of the present study was to characterise the lipid profile and distribution of these biomarkers and to estimate the prevalence of dyslipidaemia in Portugal using both the recommended values proposed by scientific societies and the percentiles for lipid metabolism biomarkers estimated specifically for the Portuguese population in this study. To the best of our knowledge, this is the most comprehensive characterisation of the lipid profile of the Portuguese population.

## 2. Materials and Methods

### 2.1. Study Population

All samples, along with the demographic and clinical data used in this study, were obtained from the e_COR study—a pre-designed and developed observational cross-sectional epidemiological study conducted by our research group. The primary objective of the e_COR study was to determine the prevalence of cardiovascular risk factors in the Portuguese population [11]. Additional objectives included sub-studies focusing on the characterisation of dyslipidaemia, as well as biochemical and genetics studies. The e_COR study included 1688 unrelated adults (98% Caucasians), comprising 848 men and 840 women aged 18 to 79 years, from the five continental regions of Portugal: Norte, Centro, Lisboa, Alentejo, and Algarve. All data were collected between January 2012 and December 2014 by specialised laboratory technicians and/or nurses using a comprehensive questionnaire, blood collection, and clinical measurements. The e_COR study was approved by the National Data Collection Commission and National Institute of Health (INSA) Ethics Committee, with participants providing informed consent for each aspect of the study. This project is a sub-study of the e_COR study, involving an extensive characterisation of one of the major cardiovascular risk factors, dyslipidaemia.

### 2.2. Biochemical Analysis

For each participant, 12 h fasting blood samples were collected. Biochemical analyses were performed at INSA’s central laboratory. Tests for TC, direct LDL-C, HDL-C, TG, apoA1, apoB, and Lp(a) were conducted on all 1688 samples using an autoanalyser Cobas Integra 400 plus (Roche, Risch-Rotkreuz, Switzerland), through enzymatic colorimetric or immunoturbidimetric methods. Serum levels of small dense low-density lipoprotein cholesterol (sdLDL-C) were measured in 1674 samples (99.2%) via direct quantification using an enzymatic colorimetric method (sLDL-EX “Seiken”) and an autoanalyser RX Daytona (Randox Laboratories, Crumlin, United Kingdom). The Lp(a) method has a detection limit of 20 nmol/L; for percentile estimation, this value was considered for individuals with values ≤20 nmol/L, but such values are not presented in the percentile tables (indicated as ‘-’). Non-HDL-C and VLDL levels were calculated as previously described [12,13]: non-HDL-C = TC minus HDL-C, and VLDL = TG divided by 5.

### 2.3. Statistical Analysis

The e_COR sampling design and sample size were determined to estimate the prevalence of cardiovascular risk factors in each Portuguese region. The original design of the total sample was not representative of the Portuguese population in terms of age and sex distribution and therefore could not be used directly to estimate percentiles or prevalence of the parameters of interest. Data from the 2011 Portuguese census were used as the reference population and source for age, sex, and region distribution (NUTS II from CENSUS 2011, Instituto Nacional de Estatística) [14] for weighted calculations to obtain results representative of the mainland Portuguese population. Statistical analyses were performed using RStudio software (version 2023.06.0+421) (R: The R Project for Statistical Computing).

#### 2.3.1. Determination of Lipid and Lipoprotein Percentiles

Subjects with characteristics known to affect lipid metabolism—such as medical history of diabetes, hyperthyroidism, hypothyroidism, and use of lipid-lowering therapy—were excluded from percentile estimation calculations. Analysis of the sample sizes for each stratum (age, sex, and geographical region) revealed that the 70–79 years age group had low representativity and was therefore excluded from percentile estimation. Consequently, a total of 1011 adults (500 men and 511 women aged 18 to 69 years) were included in the determination of 5th, 10th, 25th, 50th, 75th, 90th, and 95th percentiles for each lipid biomarker: TC, LDL-C, HDL-C, TG, apoB, apoA1, non-HDL-C, VLDL, sdLDL-C, and Lp(a).

Bootstrapping was employed to address the lack of representativity in the overall sample. The approach involved sampling with replacement from the total sample, generating a high number of subsamples following a sampling scheme that respected the age and sex distribution of the reference Portuguese population across the regions. Percentiles of interest were obtained for each subsample, resulting in a large number of estimated values for each percentile. The median of each collection was used as the percentile estimate, while the 0.025 and 0.975 sample quantiles were used as the 95% confidence interval (CI) limits.

Due to the sampling design, it was necessary to assess the homogeneity among the strata induced by stratification variables before proceeding to the percentile estimation. Firstly, the homogeneity of lipid biomarker distribution among regions was tested within each age group and sex using the Kruskal–Wallis non-parametric test. For age groups showing evidence of lack of homogeneity, Dunn’s multiple comparisons test was applied to assess homogeneity among pairs of regions. Regions where the homogeneity hypothesis was not rejected were grouped and analysed as a single stratum (a *p*-value of 0.1 was considered significant for rejecting the homogeneity hypothesis). Percentiles were then estimated using bootstrapping, with data randomly resampled 50,000 times, and the number of distinct bootstrap samples determined according to stratum weights. As a result, the percentiles were estimated to be representative of the adult Portuguese population.

#### 2.3.2. Analysis of Dyslipidaemia Prevalence

For this analysis, the entire e_COR population was included. Stratified random sampling techniques were applied based on the actual population structure [15]. These techniques enabled the construction of a weighed estimator of prevalence (expressed as a percentage) and the corresponding 95% CI, with known asymptotic probabilistic behaviour leading to the calculation of these estimations. Stratum weights were calculated in each region, by sex and age, according to the demographic composition of the adult population residing in Portugal in 2011 [14].

For reference values in dyslipidaemia prevalence analysis, we used both the recommended values for low- or moderate-risk populations from scientific societies—TC < 190 mg/dL, LDL-C < 116 mg/dL, apoB < 100 mg/dL, non-HDL-C < 130 mg/dL, TG < 150 mg/dL [4], and Lp(a) < 125 nmol/L [16]—as well as the 90th and 10th percentile values estimated for the adult Portuguese population in this study. For prevalence calculation, having either a measured value above the threshold or being on statins was considered for TC, LDL-C, apoB, non-HDL-C, and sdLDL-C. Simple frequency comparisons were performed using a Pearson’s chi-squared test with Yates’ continuity correction.

### 2.4. Testing for Familial Hypercholesterolemia

Individuals with TC or LDL-C values above the 95th percentile, combined with a family history of pCVD or hypercholesterolaemia, were sequenced for *LDLR*, *APOB*, and *PCSK9* genes, as previously reported [17]. Identified variants were classified according to *LDLR*-specific interpretation guidelines [18].

### 2.5. CVD Diagnosis

In this study, CVD was defined as a diagnosis of acute myocardial infarction, stroke, transient ischemic attack, or peripheral arterial disease made by a clinician and reported by the participant or by an associated intervention (coronary angioplasty, coronary artery bypass graft surgery, or peripheral arterial intervention). Since the e_COR study was retrospective, the association between CVD and lipid profiles is biased by the management these individuals received after their CVD event, likely including the initiation of statin therapy. Consequently, odds ratio analyses for TC, LDL-C, apoB, non-HDL-C, and sdLDL-C values were performed only on individuals not receiving statins. Due to the relatively small number of CVD cases, odds ratio analyses were conducted using Fisher’s Exact Test for Count Data.

## 3. Results

### 3.1. Percentile Estimation for Lipid Metabolism Biomarkers

Percentiles for lipids and lipoproteins—namely TC, LDL-C, HDL-C, TG, apoA1, apoB, sdLDL-C, Lp(a), VLDL, and non-HDL-C—were calculated across both sexes and various age groups (18–29, 30–39, 40–49, 50–59, 60–69, and 18–69 years old). These results are summarised in Table 1, while the 95% confidence intervals (CI) are provided in Appendix A.

For the overall population, the 50th percentile (P50) values were as follows: TC = 194 mg/dL, LDL-C = 123 mg/dL, HDL-C = 54 mg/dL, TG = 88 mg/dL, Lp(a) = 36 nmol/L, apoB = 92 mg/dL, apoA1 = 150 mg/dL, non-HDL-C = 139 mg/dL, sdLDL-C = 26 mg/dL, and VLDL = 18 mg/dL. The 90th percentile (P90) values were as follows: TC = 244 mg/dL, LDL-C = 169 mg/dL, TG = 175 mg/dL, Lp(a) = 168 nmol/L, apoB = 128 mg/dL, non-HDL-C = 193 mg/dL, sdLDL-C = 47 mg/dL, and VLDL = 35 mg/dL. For HDL-C and apoA1, the 10th percentile (P10) values were 38 mg/dL and 118 mg/dL, respectively.

### 3.2. Characterisation of the Lipid Profile

All 1688 individuals were included in the characterisation of dyslipidaemia in the Portuguese population. Stratified random sampling techniques were employed for the prevalence estimation. Dyslipidaemia was assessed in two ways: (1) by identifying individuals with lipid values above those recommended by scientific societies for the general population (low or moderate risk) and (2) by identifying individuals with lipid values above the P90, except for HDL-C and apoA1, where values below P10 were considered.

In the first method, individuals were classified as having dyslipidaemia if their lipid levels were equal to or exceeded the values recommended by scientific societies for low- or moderate-risk populations. In the second method, individuals were classified as having dyslipidaemia if their lipid levels were above the P90 (or below the P10, accordingly). In both methods, being under lipid-lowering medication was also considered for TC, LDL-C, apoB, non-HDL-C, and sdLDL-C.

#### 3.2.1. Evaluation of Dyslipidaemia by Recommended Values

The prevalence of lipid and lipoprotein values equal to or above the recommended levels for low-risk or moderate-risk populations in the Portuguese population, stratified by sex and age, is presented in Figure 1. National prevalence of elevated lipid values was as follows; 64.6% for TC, 66.9% for LDL-C, 51.3% for apoB, 68.9% for non-HDL-C, 17.8% for TG, and 21.1% for Lp(a). Significant differences (*p* < 0.05) between sexes were observed for LDL-C, apoB, non-HDL-C, and TG, with men showing higher prevalence than women. Prevalence increased with age across all parameters, particularly from the 18–29 age group to the 50–59 age group, except for Lp(a), where no significant differences were noted.

##### Analysis of Lipid Values by Treatment

Among all individuals undergoing lipid-lowering therapy for hypercholesterolaemia (23.0% CI = [20.9–25.1%]), 55.9% had TC levels below 190 mg/dL; 60.4% had LDL-C levels below 116 mg/dL; 69.6% had apoB levels below 100 mg/dL; and 54.4% had non-HDL-C levels below 130 mg/dL (Table 2). The proportion of men and women receiving medication was very similar, with 22.3% (CI = [19.3–25.4%]) of men and 23.6% (CI = [20.7–26.5%]) of women undergoing treatment.

Table 2 also presents data on individuals not taking statins who had lipid levels below the recommended values for low and moderate risk. Among these individuals, a significantly higher percentage of women (*p* < 0.05) had LDL-C, apoB, and non-HDL-C levels below the recommended values compared to men.

#### 3.2.2. Evaluation of Dyslipidaemia by Percentiles

For individuals aged 70 to 79 years, the percentiles from the 60–69 age group were used as a reference. The main results are presented in Figure 2. The estimated prevalence of severe hypercholesterolaemia (values above the P90) in the Portuguese population was as follows: TC = 30.1%, LDL-C = 30.3%, apoB = 30.4%, sdLDL-C = 31.3%, non-HDL-C = 29.8%, and Lp(a) = 13.1%. The prevalence of hypertriglyceridemia (values above the P90) was 12.0%, while low levels (below P10) of HDL-C and of apoA1 were observed in 9.0% and 10.2% of individuals, respectively. No significant differences were observed between sexes, and age-related differences were primarily attributed to the prevalence of statin use.

##### Analysis of Lipid Values by Treatment

Approximately 5% of individuals in the e_COR study who were on statin therapy had lipid levels above their age- and sex-adjusted P90. Among those not on statins, 10% exhibited lipid levels above the P90 for all lipid parameters directly influenced by statins—TC, LDL-C, apoB, and non-HDL-C. The only exception was sdLDL-C, where the prevalence was 10% in both groups—those on and not on statins (Table 3A). No significant differences were observed between men and women.

For lipid parameters typically not influenced by statins or only marginally affected (Table 3B), the prevalence of individuals with values above P90 was 10% for TG and VLDL and 13% for Lp(a), while the prevalence of values below P10 was 9% for HDL and apoA1. Notably, a significantly higher percentage of women (15%) had Lp(a) levels above P90 compared to men (11%).

#### 3.2.3. Diagnosis of Monogenic Causes of Dyslipidaemia

In our sample, 33 individuals had LDL-C or TC levels above the 95th percentile and family history of pCVD or hypercholesterolaemia, warranting genetic testing for familial hypercholesterolemia (FH). Genetic analysis revealed two pathogenic variants in the *LDLR* gene among three individuals from the e_COR study: one variant in exon 4, c.670G>A/p.(Asp224Asn), and another in exon 9, c.1291G>A/p.(Ala431Thr). All three individuals were heterozygous for these variants, confirming a diagnosis of heterozygous FH. Notably, all three individuals were from the same region, Algarve. The oldest, a 73-year-old, had experienced a MI at 57 and a stroke at 65. At the time of data collection, his LDL-C was 209 mg/dL despite being on rosuvastatin. The other two individuals, aged 25 and 27, had no history of CVD events and had LDL-C levels of 203 and 192 mg/dL, respectively, without statin therapy.

#### 3.2.4. Cardiovascular Disease and Dyslipidaemia

The prevalence of CVD in the Portuguese population was estimated to be 5.3% (Appendix A), with a slightly higher rate in men (5.8%) compared to women (4.9%), though this difference was not statistically significant. CVD prevalence increased with age, rising from 0% to 14%, with a more pronounced increase after age 50.

In the e_COR study, 117 individuals (6.9%) reported having at least one CVD event. Of these, only 80 (68.4%) were on statins at the time of data collection. The European Atherosclerosis Society recommends an LDL-C target of 55 mg/dL for secondary prevention (in very-high risk patients) [4]. This target was achieved by only two individuals (1.7%), one on statins and one not on statins (LDL-C = 45 mg/dL). When considering the LDL-C target of 70 mg/dL, which was previously recommended for very-high-risk patients, but is now used for high-risk patients, only 21 individuals (17.9%) met this target.

Figure 3 illustrates the association between lipid profile and history of CVD. Individuals with LDL-C, non-HDL-C and Lp(a) levels above the recommended values at data collection had 2.4 (*p* = 0.026), 2.4 (*p* = 0.025), and 1.7 (*p* = 0.011) higher odds for having a previous CVD event, respectively. However, the sample sizes used for these analyses were small and may induce bias.

## 4. Discussion

Dyslipidaemia is one of the major cardiovascular risk factors. Here, we assessed the prevalence of dyslipidaemia biomarkers in our population using two approaches: recommended values by scientific societies and population-specific percentiles determined for the first time in this project. We established percentiles for plasma TC, LDL-C, HDL-C, TG, apoA1, apoB, sdLDL-C, Lp(a), non-HDL-C, and VLDL, in the Portuguese population.

Assessing the prevalence of atherogenic biomarkers is crucial for understanding the prevalence of dyslipidaemia. Our analysis revealed that the prevalence of biomarkers related to hypercholesterolaemia exceeding recommended values for low and moderate risk is very high in the Portuguese population, ranging from 51.3% to 68.9%. The lowest prevalence was for apoB values, and the highest was for non-HDL values. Additionally, 17.8% of the population had TG values above 150 mg/dL, and 21.1% had Lp(a) values above 125 nmol/L. The prevalence of dyslipidaemia-related biomarkers increased with age, except for Lp(a), which showed no significant age-related differences, likely due to its strong genetic determination, with levels not influenced by lifestyle changes or age [16].

Like apoB, non-HDL-C measures the concentration of total atherogenic lipoproteins in plasma and serves as a valuable marker for cardiovascular risk evaluation [19,20]. Determining non-HDL-C offers advantages, such as being less expensive and more accessible, as it is calculated by subtracting HDL-C from TC, unlike apoA1 and apoB measurements. While apoB is not available in all laboratories like LDL-C or HDL-C, it plays a causative role in atherosclerosis progression [21,22,23] and should be measured in individuals with dyslipidaemia. Additionally, apolipoprotein measurements are not significantly influenced by high TG levels or diet, allowing for determination without prior fasting [24].

The 2016 and 2019 EAS guidelines emphasised TC and LDL-C as primary parameters for assessing dyslipidaemia and treatment targets [4,12]. The latest ESC guidelines (2021) continue to target LDL-C for treatment while shifting the focus to non-HDL-C for CVD risk stratification using SCORE-2 and SCORE2-OP tables [25,26,27]. The EAS consensus statement identifies Lp(a) as an independent risk factor for cardiovascular outcomes, emphasising that high Lp(a) levels pose a risk even at very low LDL-C concentrations [16].

For dyslipidaemia, reference values depend on individual cardiovascular risk as recommended by CVD prevention guidelines [4,25], and population-specific percentiles do not replace these guidelines. However, population-specific percentiles can help better identify individuals with very low or very high lipid markers, characteristic of genetic lipid metabolism disorders as hypobetalipoproteinaemia and familial hypercholesterolaemia. We recommend using the newly determined lipid percentiles of the Portuguese population as an additional clinical tool. By definition, P50 values can be considered adequate, while values above P90 and/or below P10 may indicate risk, depending on the biomarker.

In this study, we used P95 values combined with family history to better identify individuals with potential familial hypercholesterolemia, followed by genetic testing. We identified three individuals with heterozygous FH, all previously undiagnosed. Interestingly, the variant p.(Asp224Asn), common in both Portuguese and British populations [28], was found in one British individual living in Algarve. The two prior CVD events in one of the FH cases underscore the high CVD risk associated with this condition. Identifying three individuals in a sample of 1688 aligns with the previously estimated 1:500 prevalence for heterozygous FH [29], though newer studies suggest a higher prevalence of 1:250–313 [30]. However, this study was not designed to determine the prevalence of FH in Portugal.

Until now, the only percentiles for the Portuguese population were determined for primary care users, not for the general population, and included values only for TC, LDL-C, HDL-C, and TG [31]. Our analysis provides a more accurate reflection of the Portuguese reality. All percentiles were estimated using a bootstrap methodology, a valid tool for percentile determination, taking into account sex-, age-, and geographical-region-specific stratum weights. This approach was used to overcome the limitation of the e_COR sample not being representative of the Portuguese population due to the study design. When data from a population are not representative, this strategy is recommended [9,10].

Interestingly, the P50 cut-off points determined in this study for TC, LDL-C, and non-HDL-C are very similar to the ESC/EAS recommendations for low-risk and moderate-risk populations [4]: e_LIPID TC P50 is 194 mg/dL (ESC/EAS 190 mg/dL), LDL-C P50 is 123 mg/dL (ESC/EAS 116 mg/dL), and non-HDL-C P50 is 139 mg/dL (ESC/EAS 130 mg/dL). For apoB, ESC/EAS recommends a value below 100 mg/dL, which is slightly higher than the e_LIPID P50 (92 mg/dL). As expected, TG levels, which are highly diet-depended, were very variable in our sampled population. For Lp(a), the at-risk threshold of 125 nmol/L falls between the 75th (101 nmol/L) and 90th (168 nmol/L) percentiles for the Portuguese population. Given the strong genetic determination of Lp(a) values, they remain stable throughout life [32,33,34]. Thus, comparisons of this biomarker may be biased depending on the genetic background used to define recommended values, highlighting the importance of using population-specific percentiles.

The prevalences of lipid biomarkers above the P90, or below P10 for HDL-C and apoA1, showed a similar distribution across age groups. Unlike other biomarkers, HDL-C and apoA1 did not show an increase with age. Lipid biomarkers related to hypercholesterolaemia presented the highest prevalences, around 30% for all parameters. These results highlight the high atherogenic profile of the Portuguese population, especially concerning hypercholesterolaemia—a modifiable risk factor that often responds to lifestyle changes and, if not, can be often managed with various low-cost, generic medications to reduce lipid levels and cardiovascular risk. The analysis of medicated individuals achieving target lipid values also indicates that not all available strategies are being utilised. It is urgent to increase patient adherence to medication and ensure more personalised prescriptions by clinical teams.

Overall, these results indicate that dyslipidaemia is under-managed in Portugal, which will likely result in higher future CVD rates. In fact, Portugal has shifted from a low-risk to a moderate-risk country according to the new cardiovascular risk charts [25].

Before e_LIPID, the most recent study on dyslipidaemia in the general population of Portugal was conducted in 2002 [35]. The e_LIPID study showed a slight decrease in the prevalence of hypercholesterolaemia (4–5%) compared to the 2002 national study, which reported 68.5% of the Portuguese population having TC values ≥190 mg/dL and 71% with LDL-C ≥ 115 mg/dL. This decrease does not correlate with the exponential increase in statin sales in Portugal, which rose from 4,697,659 packages in 2004 to 9,780,010 in 2012; the definite daily dose per 1000 inhabitants per day increased from 35.0 in 2004 to 96.6 in 2012, representing a 176% increase [36]). Despite 23% of the current study population being on statins, with no sex differences, our results indicate that dyslipidaemic individuals are not being adequately treated or counselled. Misinformation from the media and a lack of public awareness about this important cardiovascular risk factor significantly contribute to these outcomes. The lack of adherence to healthy lifestyles and necessary medication is a pressing public health issue that needs urgent attention.

The prevalence of CVD in the Portuguese population was estimated at 5%, increasing significantly from 0% to 14% with age. These figures support the reclassification of Portugal from a low-CVD-risk category in the 2019 EAS guidelines [4] to a moderate-CVD-risk category in the 2021 ESC guidelines [25]. Only two individuals reached the 55 mg/dL LDL-C target recommended for secondary CVD prevention, and 32% of individuals with previous CVD were not on any lipid-lowering therapy. Due to this lack of lipid management in CVD patients, preliminary associations between prior CVD and lipid profile in non-medicated individuals were possible to be analysed. We found that individuals with LDL-C, non-HDL-C, and Lp(a) values above the recommended levels for the whole population had 1.7 to 2.4 higher odds of having experienced a previous CVD event. Since e_COR was an observational cross-sectional study with no access to past lipid profiles, the lipid measurements available were taken at the time of inquiry about previous CVD events, thus potentially biasing these calculations. However, these findings align with scientific evidence that LDL-C (and consequently non-HDL-C) and Lp(a) are causally linked to development of atherosclerotic CVD [16,37,38].

Although dyslipidaemia is highly prevalent in the Portuguese population, it is a modifiable risk factor. Correct and early identification of this CVD risk factor is crucial for proper management and can significantly contribute to CVD prevention, especially when identification leads to permanent lifestyle changes. Nevertheless, some dyslipidaemias have genetic causes (monogenic), associated with an inherently elevated CVD risk, such as FH. In contrast, most mild to severe dyslipidaemias result from multiple genes with small effects (polygenic dyslipidaemias), which may interact and affect the lipid profile while being modulated by non-genetic factors [39]. Polygenic dyslipidaemias are more easily influenced by lifestyle changes and can often be managed in general practice [40,41]. On the other hand, monogenic disorders such as FH require personalised lipid-lowering therapy and are best managed in specialist lipid clinics to effectively reduce LDL-C levels and CVD risk.

## 5. Conclusions

Our results highlight that hypercholesterolemia is a neglected cardiovascular risk factor in Portugal, with more than 50% of the population with TC, LDL-C, or apoB above the recommended values for low and moderate risk. In this study, we determined for the first-time lipid percentiles for plasma TC, LDL-C, HDL-C, non-HDL-C, TG, apoA1, apoB, Lp(a), sdLDL-C, and VLDL. We believe that these percentiles can be a useful additional tool to be used in the clinic to better identify at risk individuals and for the establishment of health policies. Since, in most cases, hypercholesterolemia is a modifiable risk factor, strategies to increase adherence to changes in lifestyle habits and to increase adherence to taking medication when necessary, need to be urgently discussed.

## Figures and Tables

**Figure 1 jcm-13-06965-f001:**
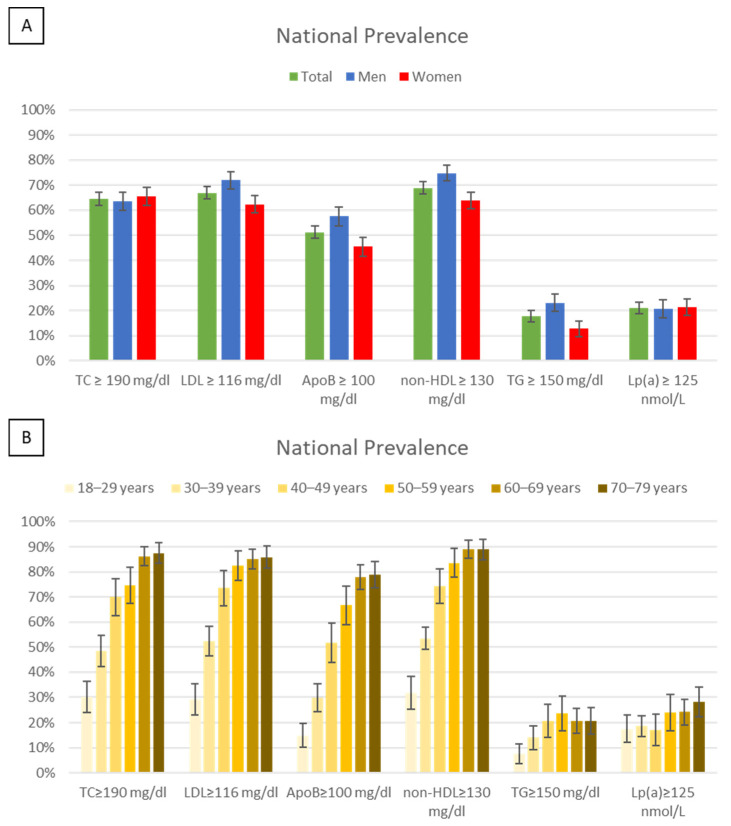
Prevalence and 95% CI of lipid parameters above the recommended values for low and moderate risk in the Portuguese population by sex (**A**) and age (**B**). ApoB, apolipoprotein B; CI, confidence interval; LDL-C, low-density lipoprotein cholesterol; Lp(a), lipoprotein (a); non-HDL-C, non-high-density lipoprotein cholesterol; TC, total cholesterol; TG, triglycerides.

**Figure 2 jcm-13-06965-f002:**
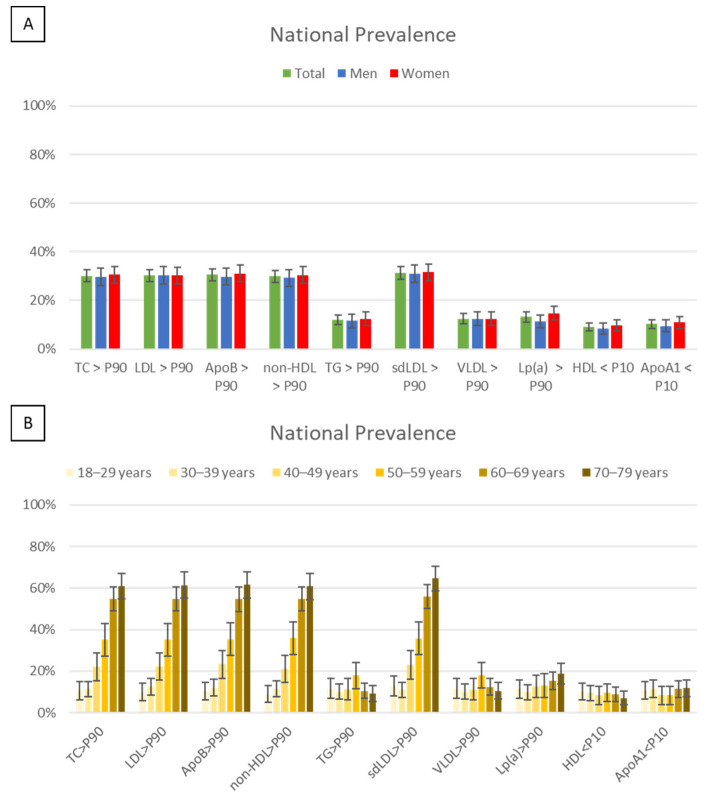
Prevalence and 95% CI of dyslipidaemia (lipid parameters above P90 or below P10) in the Portuguese population by sex (**A**) and age (**B**). ApoA1, apolipoprotein A1; ApoB, apolipoprotein B; CI, confidence interval; HDL, high-density lipoprotein cholesterol; LDL-C, low-density lipoprotein cholesterol; Lp(a), lipoprotein (a); non-HDL-C, non-high-density lipoprotein cholesterol; sdLDL-C, small dense low-density lipoprotein cholesterol; TC, total cholesterol; TG, triglycerides; VLDL, very-low-density lipoprotein cholesterol.

**Figure 3 jcm-13-06965-f003:**
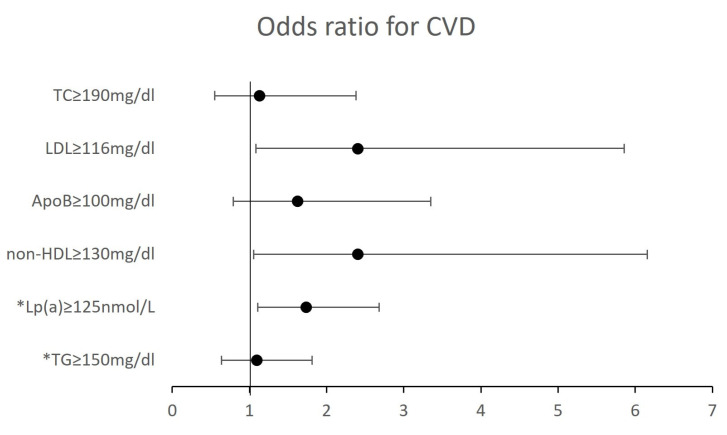
Odds ratio for having a previous CVD event, given the lipid profile. * This analysis was performed on all individuals (vs only individuals not under statins). ApoB, apolipoprotein B; LDL-C, low-density lipoprotein cholesterol; Lp(a), lipoprotein (a); non-HDL-C, non-high-density lipoprotein cholesterol; TC, total cholesterol; TG, triglycerides.

**Table 1 jcm-13-06965-t001:** Lipid and lipoprotein percentiles by sex and age group: 5th, 10th, 25th, 50th, 75th, 90th, and 95th percentiles estimated for (A) TC, (B) LDL-C, (C) HDL-C, (D) TG, (E) Lp(a), (F) apoB, (G) apoA1, (H) non-HDL-C, (I) sdLDL-C, and (J) VLDL.

(A) Total cholesterol (mg/dL)							
	5th	10th	25th	50th	75th	90th	95th
Men							
18–29 years	126	133	148	167	185	209	219
30–39 years	141	153	172	193	220	240	250
40–49 years	160	167	185	207	231	252	267
50–59 years	159	172	182	200	235	261	277
60–69 years	163	168	179	199	225	251	275
18–69 years	137	153	173	193	222	250	261
Women							
18–29 years	134	143	157	181	200	220	246
30–39 years	145	151	164	184	204	227	239
40–49 years	151	153	170	193	218	240	267
50–59 years	160	162	177	207	220	243	253
60–69 years	177	181	201	216	241	272	277
18–69 years	146	154	172	196	218	243	258
Men + Women							
18–69 years	144	153	172	194	219	244	261
(B) LDL-C (mg/dL)							
	5th	10th	25th	50th	75th	90th	95th
Men							
18–29 years	61	69	84	102	123	139	143
30–39 years	81	91	108	126	146	165	177
40–49 years	92	104	116	134	157	170	185
50–59 years	90	99	124	136	166	189	207
60–69 years	87	97	113	125	150	178	183
18–69 years	76	88	106	126	147	172	185
Women							
18–29 years	65	72	85	98	112	143	154
30–39 years	74	82	95	108	125	140	153
40–49 years	74	82	94	120	134	166	181
50–59 years	83	93	112	128	155	172	176
60–69 years	99	102	115	138	160	182	199
18–69 years	74	83	97	118	140	164	180
Men + Women							
18–69 years	74	84	101	123	144	169	183
(C) HDL (mg/dL)							
	5th	10th	25th	50th	75th	90th	95th
Men							
18–29 years	33	40	44	50	58	65	71
30–39 years	29	32	39	46	53	66	77
40–49 years	31	34	39	48	55	65	79
50–59 years	34	35	40	51	59	68	77
60–69 years	28	34	40	49	58	69	72
18–69 years	31	34	40	48	57	67	75
Women							
18–29 years	46	52	57	66	76	83	88
30–39 years	41	45	53	61	72	80	82
40–49 years	38	42	52	60	67	82	85
50–59 years	39	41	46	58	67	79	84
60–69 years	39	41	50	59	67	78	82
18–69 years	39	42	52	60	71	80	85
Men + Women							
18–69 years	34	38	45	54	64	76	83
(D) Triglycerides (mg/dL)							
	5th	10th	25th	50th	75th	90th	95th
Men							
18–29 years	45	48	57	72	99	132	159
30–39 years	49	56	71	98	141	184	230
40–49 years	56	59	80	113	165	227	294
50–59 years	52	57	68	108	127	175	241
60–69 years	57	71	87	114	146	220	253
18–69 years	50	56	69	100	136	195	246
Women							
18–29 years	39	45	60	75	101	138	155
30–39 years	41	44	59	79	100	129	158
40–49 years	43	47	54	82	106	146	188
50–59 years	51	54	69	82	104	153	178
60–69 years	49	56	72	96	129	165	189
18–69 years	43	47	63	82	107	145	178
Men + Women							
18–69 years	46	51	65	88	123	175	218
(E) Lp(a) (nmol/L)							
	5th	10th	25th	50th	75th	90th	95th
Men							
18–29 years	-	-	-	28	75	148	240
30–39 years	-	-	22	37	111	193	227
40–49 years	-	-	22	38	103	144	202
50–59 years	-	-	22	47	125	162	231
60–69 years	-	-	22	38	115	204	310
18–69 years	-	-	22	38	103	182	238
Women							
18–29 years	-	-	-	35	75	152	202
30–39 years	-	-	-	29	74	158	180
40–49 years	-	-	-	24	71	152	179
50–59 years	-	-	22	42	115	180	209
60–69 years	-	-	21	58	118	183	264
18–69 years	-	-	-	36	94	163	209
Men + Women							
18–69 years	-	-	-	36	101	168	223
(F) ApoB (mg/dL)							
	5th	10th	25th	50th	75th	90th	95th
Men							
18–29 years	49	53	63	80	91	103	107
30–39 years	62	66	82	97	113	127	134
40–49 years	65	77	91	104	116	138	143
50–59 years	72	74	93	103	125	144	151
60–69 years	72	76	90	104	121	138	147
18–69 years	60	65	81	98	113	133	144
Women							
18–29 years	52	54	67	77	89	106	117
30–39 years	57	63	71	82	95	109	118
40–49 years	61	67	75	90	100	120	135
50–59 years	66	67	83	92	110	120	128
60–69 years	73	77	87	104	119	132	147
18–69 years	57	64	75	88	104	120	131
Men + Women							
18–69 years	59	65	77	92	109	128	139
(G) ApoA1 (mg/dL)							
	5th	10th	25th	50th	75th	90th	95th
Men							
18–29 years	110	114	123	134	146	162	167
30–39 years	104	112	119	134	149	170	179
40–49 years	108	111	128	140	158	170	208
50–59 years	115	116	128	144	162	183	185
60–69 years	111	117	131	148	162	177	187
18–69 years	106	114	124	140	157	173	185
Women							
18–29 years	123	132	149	168	191	206	222
30–39 years	120	131	148	164	186	203	217
40–49 years	113	120	140	156	184	205	229
50–59 years	123	129	135	150	175	204	211
60–69 years	126	134	145	162	179	188	197
18–69 years	118	128	142	160	182	202	216
Men + Women							
18–69 years	111	118	132	150	170	194	208
(H) non-HDL-C (mg/dL)							
	5th	10th	25th	50th	75th	90th	95th
Men							
18–29 years	73	78	95	118	139	155	163
30–39 years	91	106	125	143	172	195	202
40–49 years	108	117	138	157	180	199	226
50–59 years	102	111	138	152	195	214	236
60–69 years	104	108	135	147	174	206	216
18–69 years	87	100	121	144	170	199	217
Women							
18–29 years	76	84	98	110	133	159	171
30–39 years	83	90	106	122	144	159	176
40–49 years	88	92	107	134	151	182	205
50–59 years	99	104	122	146	163	181	192
60–69 years	111	116	131	158	185	209	226
18–69 years	86	92	110	134	158	184	198
Men + Women							
18–69 years	86	96	114	139	163	193	211
(I) sdLDL-C (mg/dL)							
	5th	10th	25th	50th	75th	90th	95th
Men							
18–29 years	9	12	15	19	27	32	36
30–39 years	15	17	22	29	39	51	63
40–49 years	18	20	26	35	44	58	66
50–59 years	16	21	25	36	53	63	72
60–69 years	16	19	24	31	43	54	56
18–69 years	13	16	21	29	41	56	64
Women							
18–29 years	9	10	15	20	27	36	41
30–39 years	10	13	17	22	29	39	42
40–49 years	13	15	18	23	30	42	52
50–59 years	15	16	20	26	32	40	40
60–69 years	17	19	22	27	35	42	49
18–69 years	11	14	18	24	31	41	44
Men + Women							
18–69 years	12	15	19	26	36	47	57
(J) VLDL (mg/dL)							
	5th	10th	25th	50th	75th	90th	95th
Men							
18–29 years	9	10	11	14	20	26	32
30–39 years	10	11	14	20	28	37	46
40–49 years	11	12	16	23	33	45	59
50–59 years	10	11	14	22	25	35	48
60–69 years	9	12	16	22	28	39	45
18–69 years	10	11	14	20	27	38	48
Women							
18–29 years	8	9	12	15	20	28	31
30–39 years	8	9	12	16	20	26	32
40–49 years	9	9	11	16	21	29	38
50–59 years	10	11	14	16	21	31	36
60–69 years	10	11	14	19	26	33	38
18–69 years	9	9	13	16	21	29	36
Men + Women							
18–69 years	9	10	13	18	24	35	43

For cells with ‘-’, it was not possible to calculate the percentile due to Lp(a) detection limit, but it would be ≤20 nmol/L. TC, total cholesterol; LDL-C, low-density lipoprotein cholesterol; HDL-C, high-density lipoprotein cholesterol; TG, triglycerides; Lp(a), lipoprotein (a); apoB, apolipoprotein B; apoA1, apolipoprotein A1; non-HDL-C, total cholesterol minus high-density lipoprotein cholesterol; sdLDL-C, small dense low-density lipoprotein cholesterol; VLDL, very-low-density lipoprotein cholesterol.

**Table 2 jcm-13-06965-t002:** Frequency of individuals with lipid values below the recommended by scientific societies for low or moderate risk in e_COR study by sex and medication.

	Under Statins				Not Under Statins				Statins vs. No Statins * (*p*-Value)
	Men	Women	Total	Men vs. Women * (*p*-Value)	Men	Women	Total	Men vs. Women * (*p*-Value)	
n	228	226	454		620	614	1234		
TC < 190 mg/dL	137 (60.1%)	117 (51.8%)	254 (55.9%)	0.091	205 (44.2%)	291 (45.0%)	550 (44.6%)	0.833	**<<0.001**
LDL-C < 116 mg/dL	141 (61.8%)	133 (58.8%)	274 (60.4%)	0.578	230 (37.1%)	312 (50.8%)	542 (43.9%)	**<<0.001**	**<<0.001**
ApoB < 100 mg/dL	153 (67.1%)	163 (72.1%)	316 (69.6%)	0.289	325 (52.4%)	435 (70.8%)	760 (61.6%)	**<<0.001**	**0.003**
non-HDL-C < 130 mg/dL	127 (55.7%)	120 (53.1%)	247 (54.4%)	0.644	205 (33.1%)	291 (47.4%)	496 (40.2%)	**<<0.001**	**<<0.001**

* chi-squared test. *p*-values < 1 × 10^−5^ are represented as *p* << 0.001. *p*-values < 0.05 (statistically significant) are represented in bold. ApoB, apolipoprotein B; LDL-C, low-density lipoprotein cholesterol; n, number; non-HDL-C, non-high-density lipoprotein cholesterol; TC, total cholesterol.

**Table 3 jcm-13-06965-t003:** Frequency of individuals in e_COR study with dyslipidaemia (defined by lipid values above P90 or below P10) in e_COR study, by sex and medication, for parameters influenced (A) and not influenced (B) by statins.

A	Under Statins				Not Under Statins				
	Men	Women	Total	Men vs. Women * (*p*-Value)	Men	Women	Total	Men vs. Women * (*p*-Value)	Statins vs. No Statins * (*p*-Value)
n	228	226	454		620	614	1234		
TC > P90	14 (6.1%)	13 (5.8%)	27 (5.9%)	1	59 (10.3%)	56 (9.8%)	124 (10%)	0.820	**0.012**
LDL-C > P90	13 (5.7%)	10 (4.4%)	23 (5.1%)	0.685	66 (10.6%)	54 (8.8%)	120 (9.7%)	0.317	**0.003**
ApoB > P90	12 (5.3%)	14 (6.2%)	26 (5.7%)	0.822	58 (9.4%)	58 (9.4%)	116 (9.4%)	1	**0.021**
non-HDL-C > P90	11 (4.8%)	11 (4.9%)	22 (4.8%)	1	59 (9.5%)	56 (9.1%)	115 (9.3%)	0.888	**0.004**
sdLDL-C > P90	21 (9.2%)	23 (10.2%)	44 (9.7%)	0.850	65 (10.5%)	71 (11.6%)	136 (11%)	0.592	0.489
**B**	**Men**			**Women**		**Total**		**Men vs. Women * (*p*-value)**	
n	848			840		1688			
TG > P90	84 (9.9%)			95 (11.3%)		179 (10.6%)		0.391	
VLDL > P90	67 (10.8%)			65 (10.6%)		132 (10.7%)		0.974	
Lp(a) > P90	89 (10.5%)			124 (14.8%)		213 (12.6%)		**0.010**	
HDL < P10	63 (7.4%)			80 (9.5%)		143 (8.5%)		0.145	
ApoA1 < P10	71 (8.4%)			84 (10%)		155 (9.2%)		0.283	

* chi-squared test. *p*-values < 0.05 (statistically significant) are represented in bold. ApoA1, apolipoprotein A1; ApoB, apolipoprotein B; HDL, high-density lipoprotein cholesterol; LDL-C, low-density lipoprotein cholesterol; Lp(a), lipoprotein (a); n, number; non-HDL-C, non-high-density lipoprotein cholesterol; sdLDL-C, small dense low-density lipoprotein cholesterol; TC, total cholesterol; TG, triglycerides; VLDL, very-low-density lipoprotein cholesterol.

## Data Availability

The data presented in this study are available on request from the corresponding author due to privacy reasons.

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
