# Peer review of "Portuguese Lipid Study (e_LIPID)"

_jcm, 2024, doi:10.3390/jcm13226965_

Round 1

Reviewer 1 Report

Comments and Suggestions for Authors

„e_LIPID” in the title is misleading as it suggests this paper is related to an online project, but it is not.

The text should be corrected by a native speaker so that all wording is precise. Thus, for example: “Epidemiological studies have linked CVD to increasing values in plasma lipids” in my opinion not to “increasing” but to “elevated, and  I propose “ Epidemiological studies have linked CVD to elevated plasma lipid levels”.

“Biomarker per-centiles are of extreme importance for the definition of reference intervals for health and disease” -  Knowing the distribution of lipid biomarkers in the population is important, but it is hard to accept this statement as we have to know the relation between biomarkers and CVD risk. To my knowledge, we define dyslipidemia by establishing cutoff points, not by exceeding a certain percentile of concentrations of a particular marker.

“Dyslipidaemia characterization” - This is an incorrect title. This section does not describe the characteristics of dyslipidemia but rather refers to the diagnosis and cutoff points of the various markers.

“CVD analysis” – same as above, this is an incorrect title. This section is related to CVD diagnosis

 “Reference values for lipid metabolism biomarkers” – In my opinion, there is a distribution of biomarkers. It is hard to accept there are reference values.

“Analysis of lipid values under control versus treatment” – it is hard to understand what it means, “under” what kind of “control”

“lipid values below the recommended lipid values (line 233), eg. TC< 190 mg/dl” AS I know total cholesterol concentrations below 190 mg/dl are recognized as proper, therefore such concentrations are not below the recommended levels

“Evaluation of dyslipidemia by percentiles” – What does it mean?  In this section frequency of serum lipid concentration below 10Th and above 90th percentiles are presented.

“Analysis of dyslipidemia versus treatment” - it is hard to understand what it means.

„Possible monogenic cause of dyslipidaemias” – In this section possible monogenic causes of dyslipidemia are not described. In this section lipid disorders diagnosed in subjects with LDL-C or TC above the 95th percentile are described.

“However, using these percentile values as a reference would allow the clinician to interpret how deviated an individual value is from the expected in the global population” – It is hard to agree. The rules for diagnosing dyslipidemia and assessing CVD risk are clearly defined and do not depend on the prevalence of these disorders in the population. If different rules are adopted in Portugal than in other European countries, this should be clearly stated.

In European populations, the distribution of lipid concentrations, such as cholesterol, can be considered a normal distribution; however, the distribution of Lp(a) concentrations in European populations is a skewed distribution, and in most populations concentrations higher than 30 mg/dl are recognized as related to enhanced CVD risk. If we convert Lp(a) concentrations from nmol/L to mg/dl according to formula Lp(a) [nmol/L] = 2,5 x Lp(a) [mg/dl], we see that the 50th percentile of Lp(a) concentrations in the Portuguese population is significantly higher than 30 mg/dl. Does this mean that 50% of the population has such high concentrations?  Should the distribution of Lp(a) concentrations in Portugal be considered normal?  This needs to be clarified and discussed.

Comments on the Quality of English Language

The paper should be corrected by a native speaker familiar with medical sciences

Author Response

R1: „e_LIPID” in the title is misleading as it suggests this paper is related to an online project, but it is not.

A1: The e_LIPID study derives from the e_COR study, where the ‘e’ in Portuguese comes from the word estudo, meaning study.

R2: The text should be corrected by a native speaker so that all wording is precise. Thus, for example: “Epidemiological studies have linked CVD to increasing values in plasma lipids” in my opinion not to “increasing” but to “elevated, and  I propose “ Epidemiological studies have linked CVD to elevated plasma lipid levels”.

A2: We have revised the manuscript and done several edits to improve the language. All changes are highlighted with track changes.

R3: “Biomarker per-centiles are of extreme importance for the definition of reference intervals for health and disease” -  Knowing the distribution of lipid biomarkers in the population is important, but it is hard to accept this statement as we have to know the relation between biomarkers and CVD risk. To my knowledge, we define dyslipidemia by establishing cutoff points, not by exceeding a certain percentile of concentrations of a particular marker.

A3: We do know the relationship between certain biomarkers, as LDL, ApoB, and Lp(a) with CVD risk. Taken from the 2021 ESC guidelines on CVD: “The causal role of LDL-C, and other apo-B-containing lipoproteins, in the development of ASCVD is demonstrated beyond any doubt by genetic, observational, and interventional studies.”, and from the Lp(a) EAS consensus statement: “Epidemiologic and genetic studies involving hundreds of thousands of individuals strongly support a causal and continuous association between Lp(a) concentration and cardiovascular outcomes in different ethnicities; elevated Lp(a) is a risk factor even at very low levels of low-density lipoprotein cholesterol.”

And while most dyslipidemia definitions at the individual level do depend on cut-off points, even the Dutch lipid clinical network score has ‘First-degree relative with known LDL cholesterol >95th percentile by age and gender’ as criteria, highlight the importance of having national level age and sex-adjusted percentiles.

“For dyslipidaemia the reference values for each person depend on their cardiovascular risk as recommended in CVD prevention guidelines [4,23]. However, using percentile values as a reference would allow the clinician to interpret how deviated an individual value is from the expected in the global population. Moreover, percentiles can help to determine which individuals should be further studied due to very low or very high lipid markers characteristic of genetic diseases of lipid metabolism as hypobetalipoproteinaemia and familial hypercholesterolaemia.”

R4: “Dyslipidaemia characterization” - This is an incorrect title. This section does not describe the characteristics of dyslipidemia but rather refers to the diagnosis and cutoff points of the various markers.

“CVD analysis” – same as above, this is an incorrect title. This section is related to CVD diagnosis

 “Reference values for lipid metabolism biomarkers” – In my opinion, there is a distribution of biomarkers. It is hard to accept there are reference values.

A4: We have changed these titles according to the reviewer suggestions, they now read “2.5 Dyslipidaemia prevalence analysis”, “2.7 CVD diagnosis” and “3.1 Percentile estimation for lipid metabolism biomarkers”

R5: “Analysis of lipid values under control versus treatment” – it is hard to understand what it means, “under” what kind of “control”

“Analysis of dyslipidemia versus treatment” - it is hard to understand what it means.

A5: Thank you, we have changed the structure of the paragraphs to be clearer what were the analysis performed in each paragraph. We did mention that we meant ‘under control’ as individuals that had lipid values below the recommended values for the general population, but we have replaced any mention of under control to say only ‘under recommended values’ to avoid confusion.

R6: “lipid values below the recommended lipid values (line 233), eg. TC< 190 mg/dl” AS I know total cholesterol concentrations below 190 mg/dl are recognized as proper, therefore such concentrations are not below the recommended levels

A6: The recommended values are indeed 190mg/dl for TC, so individuals under 190mg/dl, have lipid values below the ‘recommended values’.

R7: “Evaluation of dyslipidemia by percentiles” – What does it mean?  In this section frequency of serum lipid concentration below 10Th and above 90th percentiles are presented.

A7: Thank you for the opportunity to elaborate on this issue. We have stated that we would be analysing the prevalence of dyslipidaemia in the Portuguese population by 2 ways: either by the cut-off points according to global cardiovascular risk, meaning an individual with lipid values above the recommended cut-offs would be considered as having dyslipidaemia; or by the sex and age-adjusted percentiles, meaning that and individual having lipid values above the 90th percentile (or below the 10th percentile for HDL and ApoA1), would be considered as having dyslipidaemia.

We have expanded this explanation in paragraph 3.2.

R8: „Possible monogenic cause of dyslipidaemias” – In this section possible monogenic causes of dyslipidemia are not described. In this section lipid disorders diagnosed in subjects with LDL-C or TC above the 95th percentile are described.

A8: We have changed this title to “Diagnosis of monogenic cause of dyslipidaemias”

R9: “However, using these percentile values as a reference would allow the clinician to interpret how deviated an individual value is from the expected in the global population” – It is hard to agree. The rules for diagnosing dyslipidemia and assessing CVD risk are clearly defined and do not depend on the prevalence of these disorders in the population. If different rules are adopted in Portugal than in other European countries, this should be clearly stated.

A9: The rules for CVD risk are the same in Portugal and in Europe, as we mention in line 347 (just before this phrase) in the discussion. In here we just state that percentiles are a good tool to help interpret the lipid profile of an individual as compared to the general population. Additionally, the 90th or 95th percentile are helpful, as they can indicate individuals to be further studied for genetic diseases of lipid metabolism as familial hypercholesterolaemia, for example.

R10: In European populations, the distribution of lipid concentrations, such as cholesterol, can be considered a normal distribution; however, the distribution of Lp(a) concentrations in European populations is a skewed distribution, and in most populations concentrations higher than 30 mg/dl are recognized as related to enhanced CVD risk. If we convert Lp(a) concentrations from nmol/L to mg/dl according to formula Lp(a) [nmol/L] = 2,5 x Lp(a) [mg/dl], we see that the 50th percentile of Lp(a) concentrations in the Portuguese population is significantly higher than 30 mg/dl. Does this mean that 50% of the population has such high concentrations?  Should the distribution of Lp(a) concentrations in Portugal be considered normal?  This needs to be clarified and discussed.

A10: Thank you for this comment. We used in the work the cut-off of 125nmol/L (loosely corresponding to 50mg/dl) according to the 2021 EAS consensus statement “Rather than absolute values, clinical guidelines should consider using risk thresholds with ‘grey’ zones (e.g., 30–50 mg/dL or 75–125 nmol/L) to either rule-in (≥50 mg/dL; 125 nmol/L) or rule-out (<30 mg/dL; 75 nmol/L) cardiovascular risk.” With this cut-off, we have 20% of the Portuguese population with high Lp(a) values. Additionally, this consensus statement highlights the differences in Lp(a) concentrations between different ethnicities, so population specific percentiles are important to understand how deviated from the general population an individual’s Lp(a) value really is.

Reviewer 2 Report

Comments and Suggestions for Authors

I have received for review a commentary entitled “e_LIPID – CHARACTERISATION OF THE LIPID PROFILE 2 OF THE PORTUGUESE POPULATION” which is being processed by the journal Journal of Clinical Medicine.

I would like to congratulate the collective of authors for the proposed manuscript. The proposed case is an extremely interesting one, with therapeutic and prognostic value.  Authors should pay attention to the following aspects in order to improve the proposed manuscript:

Introduction - provides a comprehensive review of the theme of the manuscript and the reasons for the choice.

Materials and method - a section on statistical analysis should be included

The table should be numbered and further explanation of abbreviations is required

Results - the descriptive study is extremely interesting because of the large number of patients evaluated and the important public health impact. I suggest the authors to introduce additional correlations (risk scores, paraclinical parameters - electrocardiographic, echocardiographic, associated with subclinical atherosclerosis).

Discussion - presents the main results obtained by comparison with other studies previously published in the literature.

Conclusions - captures the main issues highlighted in this study

In conclusion, the proposed manuscript brings to attention an extremely interesting topic, presenting scientific information with therapeutic value. The quality of the manuscript will be improved if the authors take into account the remarks made above.

Author Response

The reviewer mentions that a statistical analysis section should be included, but we do indeed have a statistical analysis section in the methods, “2.3. General statistical analysis”. Additionally, in sections 2.4, 2.5 and 2.6 we specify the statistical methods used for determination of percentile values and any statistical tests used.

The tables are numbered, but table 1 is indeed composed of several smaller tables. We have added letters to each individual table to help in referencing.

In refence to the suggestion about other correlations, the objective of this study was to ‘characterise the lipid profile and the distribution of these biomarkers, and to estimate the prevalence of dyslipidaemia in Portugal’. Having clinical parameters as ECG, imaging or other measures of subclinical atherosclerosis, although an interesting addition for future studies, was outside the scope of the present study.

Round 2

Reviewer 1 Report

Comments and Suggestions for Authors

Title - I would like to point out that for me, as for most readers, e-Lipid means as much as e-mail. Most readers do not speak Portuguese. When you decide to publish your work in an international journal, you must adapt to how the international community thinks and understands.

I understand that you are working on the text and grammar, but I still strongly recommend that you work with a native speaker who knows medical vocabulary and scientific speaking style. In my opinion, the paper is written in what I would call "rather harsh" language.

In the authors' response to my comments, I found sentences that I disagree with. It is not a correct interpretation that we diagnose FH when LDL-chol > 95 percentaile[‘First-degree relative with known LDL cholesterol >95th percentile by age and gender’ as criteria, highlighting the importance of having national-level age and sex-adjusted percentiles]. If we have such a high LDL-chol in the family of a patient with FH, we can be almost certain that this family member will also have FH. And that's just it. It is quite common to find lower cholesterol levels in FH patients than >95 percentaile. Whenever we have a family member of an FH patient with significantly elevated cholesterol and LDL cholesterol levels, we must consider the presence of FH. The expression of markers depends on many factors, including the type of molecular defect.

I do not accept the authors' interpretation of recommended concentrations of cholesterol, LDL cholesterol, etc. Scientific societies define cut-off values, i.e. these values should be interpreted as recommendations above which we either diagnose disorders or define as associated with high risk. Therefore, cholesterol concentration of 190 mg/dL is not the recommended concentration but the recommended cut-off value.  When interpreting a patient's test, we use this cutoff and consider cholesterol concentrations below 190 mg/dL as normal/proper/etc. This is the common interpretation [to my knowledge, and I have been involved in these issues for over thirty years]. Therefore, I do not accept the authors' interpretation and would like to draw your attention to the need to make appropriate corrections.

"Under control" means that something is under control, i.e. that we control it well by taking certain actions, not that we are dealing with a correct process and that the markers describing it have correct values. The language must be precise and understandable to the reader. Therefore, I recommend consulting a native speaker who works in this field and is familiar with the wording used by professionals.  The authors must change this term, as it does not describe reality correctly. It is clear from the text that we are dealing with normal values in people without metabolic disorders.

I also recommend that you re-read my previous recommendations and suggestions and make appropriate corrections throughout the text to make it understandable to everyone, not just the Portuguese.

Comments on the Quality of English Language

require attention and modification

Author Response

Comments 1 - Title - I would like to point out that for me, as for most readers, e-Lipid means as much as e-mail. Most readers do not speak Portuguese. When you decide to publish your work in an international journal, you must adapt to how the international community thinks and understands.

Reply 1 – As requested, we have changed the title. We still maintain the e_LIPID designation in the text because we need to distinguish this sub-study (e_LIPID) from the main previous study (e_COR), but hopefully we decreased its relevance to avoid confusion with an electrical component.   

Comments 2 - I understand that you are working on the text and grammar, but I still strongly recommend that you work with a native speaker who knows medical vocabulary and scientific speaking style. In my opinion, the paper is written in what I would call "rather harsh" language.

Reply 2 – We have revised extensively the text using AI suggestions, which were all reviewed to ensure the scientific rigor and coherence were maintained. Hopefully we have softened the language.

Comments 3 - In the authors' response to my comments, I found sentences that I disagree with. It is not a correct interpretation that we diagnose FH when LDL-chol > 95 percentaile[‘First-degree relative with known LDL cholesterol >95th percentile by age and gender’ as criteria, highlighting the importance of having national-level age and sex-adjusted percentiles]. If we have such a high LDL-chol in the family of a patient with FH, we can be almost certain that this family member will also have FH. And that's just it. It is quite common to find lower cholesterol levels in FH patients than >95 percentaile. Whenever we have a family member of an FH patient with significantly elevated cholesterol and LDL cholesterol levels, we must consider the presence of FH. The expression of markers depends on many factors, including the type of molecular defect.

I do not accept the authors' interpretation of recommended concentrations of cholesterol, LDL cholesterol, etc. Scientific societies define cut-off values, i.e. these values should be interpreted as recommendations above which we either diagnose disorders or define as associated with high risk. Therefore, cholesterol concentration of 190 mg/dL is not the recommended concentration but the recommended cut-off value.  When interpreting a patient's test, we use this cutoff and consider cholesterol concentrations below 190 mg/dL as normal/proper/etc. This is the common interpretation [to my knowledge, and I have been involved in these issues for over thirty years]. Therefore, I do not accept the authors' interpretation and would like to draw your attention to the need to make appropriate corrections.

Reply 3 – As requested we have reviewed the whole manuscript, mainly the discussion text and structure to lessen the relevance placed on percentiles as a diagnostic tool but instead to use percentiles as an additional clinical tool, to help better identify individuals with very low or very high lipid markers, which might have inherent genetic disorders that lead to those high or low levels. Text about recommended cut-off values was also changed.

Comments 4 - "Under control" means that something is under control, i.e. that we control it well by taking certain actions, not that we are dealing with a correct process and that the markers describing it have correct values. The language must be precise and understandable to the reader. Therefore, I recommend consulting a native speaker who works in this field and is familiar with the wording used by professionals.  The authors must change this term, as it does not describe reality correctly. It is clear from the text that we are dealing with normal values in people without metabolic disorders.

Reply 4 – We have removed the term ‘under control’ from the text and replaced it with ‘having lipid levels below the recommended values’.

Comments 5 - I also recommend that you re-read my previous recommendations and suggestions and make appropriate corrections throughout the text to make it understandable to everyone, not just the Portuguese.

Reply 5 – We have revised the whole manuscript as requested and hope that it is now understandable to everyone.